# Anti-Inflammatory Diet and Protein-Enriched Diet Can Reduce the Risk of Cognitive Impairment among Older Adults: A Nationwide Cross-Sectional Research

**DOI:** 10.3390/nu16091333

**Published:** 2024-04-28

**Authors:** Liang Wang, Xiaobing Xian, Mengting Zhou, Ke Xu, Shiwei Cao, Jingyu Cheng, Weizhi Dai, Wenjia Zhang, Mengliang Ye

**Affiliations:** 1School of Public Health, Chongqing Medical University, Chongqing 400016, China; 2022120779@stu.cqmu.edu.cn (L.W.); xiaobing@stu.cqmu.edu.cn (X.X.); 2023222360@stu.cqmu.edu.cn (M.Z.); xk2548697188@outlook.com (K.X.); 2022222049@stu.cqmu.edu.cn (J.C.); 2023222358@stu.cqmu.edu.cn (W.Z.); 2School of the Second Clinical, Chongqing Medical University, Chongqing 400016, China; 2021220723@stu.cqmu.edu.cn; 3School of the First Clinical, Chongqing Medical University, Chongqing 400016, China; 2022221121@stu.cqmu.edu.cn

**Keywords:** anti-inflammatory diet, cognitive impairment, older adults, protein-enriched diet

## Abstract

Background: Cognitive impairment (CI) is a common mental health disorder among older adults, and dietary patterns have an impact on cognitive function. However, no systematic researches have constructed anti-inflammatory diet (AID) and protein-enriched diet (PED) to explore their association with CI among older adults in China. Methods: The data used in this study were obtained from the 2018 waves of the China Longitudinal Health and Longevity Survey (CLHLS). We construct AID, PED, and calculate scores for CI. We use binary logistic regression to explore the relationship between them, and use restrictive cubic splines to determine whether the relationships are non-linear. Subgroup analysis and sensitivity analysis were used to demonstrate the robustness of the results. Results: A total of 8692 participants (mean age is 83.53 years) were included in the analysis. We found that participants with a higher AID (OR = 0.789, 95% confidence interval: 0.740–0.842, *p* < 0.001) and PED (OR = 0.910, 95% confidence interval: 0.866–0.956, *p* < 0.001) score showed lower odds of suffering from CI. Besides, the relationship between the two dietary patterns and CI is linear, and the results of subgroup analysis and sensitivity analysis are also significant. Conclusion: Higher intakes of AID and PED are associated with a lower risk of CI among older adults, which has important implications for future prevention and control of CI from a dietary and nutritional perspective.

## 1. Introduction

With the rapid growth of China’s aging population and longer life expectancy, age-related disorders have become a major public health problem [1]. One of the worrying issues is the deterioration of cognitive function (CF) [2], which may not only seriously harm individuals’ well-being, but also impose a huge burden on their families and caregivers, as well as the financial and health systems of the society [3]. CF declines with age [4], age-related cognitive impairment (CI) is a common disorder in older adults, ranging from mild CI to severe dementia [5], and with the aging of the global population, dementia is an increasingly serious public health issue [6]. It is expected that the number of dementia patients will reach 82 million by 2030 and 152 million by 2050 as the population ages [7]. The cost of caring for dementia patients worldwide exceeds USD 800 billion annually and is expected to increase to USD 2 trillion by 2030 [8]. China has the largest number of dementia patients in the world (9.5 million), followed by the United States (4.2 million) [9]. According to a nationwide survey conducted in China from 2015 to 2018, approximately 15.5% of the population aged 60 and above suffered from mild CI, 6.0% suffered from dementia, and 3.9% had Alzheimer’s disease [10]. Due to the incurable nature of dementia, it is crucial to detect CI in the early stages, and maintaining CF is a core component of successful aging [11].

The Lancet Commission concluded that lifestyle factors, including diet and nutrition, could prevent or delay 40% of dementia worldwide [12]. For instance, in terms of physical activity, a study has found that an increase in exercise and resistance exercises are beneficial for CF [13], and they can reduce the risk for dementia [14]. What is more, a combination of aerobic and strength training can slow the decline in motor and cognitive abilities in people with dementia [15]. In the aspects of nutrition and diet, a study proved that beta-carotene is a powerful antioxidant and dietary precursor of vitamin A, playing an important role in maintaining mental health and CF [16]. Moreover, dietary polyphenols have a neuroprotective effect and may prevent or improve CI [17], and there is emerging evidence to suggest that several plant-based foods rich in polyphenols have a promoting effect on brain health [18]. Omega-3 fatty acids, found in some foods such as fish, legumes, and nuts, also have an impact on CI [19]; Omega-3 fatty acids include eicosapentaenoic acid (EPA) and docosahexaenoic acid (DHA) [20], which are an important factor affecting the health of the brain structure and function [21]. Therefore, implementing a healthy eating pattern is a potential strategy for preventing CI, which can alleviate CI that occurs with age [22]. As is well known, there is a certain relationship between some nutrients in the diet and inflammation [23]. The Dietary Inflammatory Index (DII) is a comprehensive indicator calculated based on the correlation between nutrients and systemic pro-inflammatory cytokine levels, which has the function of evaluating the likelihood of dietary inflammation [24]. Higher DII scores illustrate more pro-inflammatory diets, while more negative values indicate more anti-inflammatory diets [25]. The characteristic of aging is an increase in the concentration of many pro-inflammatory molecules in the circulation, a phenomenon known as “inflammation” [26,27]. Systemic inflammation can lead to cognitive decline and dementia by inducing a reactive pro-inflammatory environment in the central nervous system [28], and research has proved that higher pro-inflammatory dietary potential is associated with increased incidence of dementia [29]. Simultaneously, an anti-inflammatory diet (AID) may have an impact on some mental health disorders. Previous studies have found that AIDs can serve as potential interventions for depression [30], and adopting the interventions of anti-inflammatory and diversified diets may alleviate the burden of depression among older adults [31].

Earlier studies have found that many dietary patterns are associated with improved CF. A *JAMA* article showed that adding olive oil or nuts to the Mediterranean diet was associated with improvements in CF in older adults [32], and a number of other related studies have further confirmed the significant association between the Mediterranean diet and CF [33,34,35]. Additionally, the Mediterranean-DASH Intervention for Neurodegenerative Delay, known as the MIND diet, is a hybrid of the Mediterranean diet and the DASH (Dietary Approaches to Stop Hypertension) diet, which can prevent and slow the cognitive decline in older persons [36,37,38,39]. A prospective cohort study found that AID patterns are associated with decreased grip strength [40], and research has found that grip strength is associated with cognition and dementia [41,42], so there may be a certain connection between AID and CI. In the elderly population in northern China, a pro-inflammatory diet is associated with an increased risk of mild CI [43]. Despite numerous studies on other dietary patterns and CI, as well as studies on AID and other outcomes variables, research on the relationship between AID and CI among older Chinese adults remains undefined, which requires further research and certification.

Protein is a critical nutrient for normal CF [44], as some research suggested that protein intake is associated with better CF performance [45,46]. Creatine is found in many protein-rich meats, such as beef, lamb, chicken, and fish, and studies have found that creatine has a positive effect on brain health [47] and cognitive performance [48]. This may also be one of the reasons why protein-rich diets have a positive effect on CF. A balanced protein diet is beneficial for the CF of older adults in Japan [49]. Besides, a study has found that the dietary protein intake, total animal protein intake, total meat, egg, and legume protein intake of adults aged 60 and above in foreign countries are positively correlated with CF [50]. Studies have shown that a protein-enriched Mediterranean diet fights malnutrition and promotes healthy neurocognitive aging in older adults [51]. Moreover, fish products are recommended as a protein dietary source and are associated with a lower risk of CI [52], while milk and dairy intake in middle age may have a protective effect against CI [53]. Nonetheless, other studies reported that there is no significant correlation between protein intake and CF [54,55]. Considering that previous research on the relationship between protein intake and CF were not entirely consistent, and no one had built a systematic protein-enriched diet (PED) to explore its relationship with CI, we further explored the association between them among older adults in China by using a systematic dietary pattern.

Based on the analysis above, since the role of protein and anti-inflammatory diets on CI is unclear, there is an increasing need to construct two dietary indices for measuring their relationship with CI. Therefore, in this study, we will construct two dietary indices, AID and PED, and explore their effects on CI among Chinese older adults based on the CLHLS database.

## 2. Materials and Methods

### 2.1. Participants and Process

The data used in this study were obtained from the 2018 waves of the China Longitudinal Health and Longevity Survey (CLHLS), which used a multi-stage stratified cluster sampling design in 23 out of 31 provinces in China. In this research, we used the data of the 2018 wave to investigate the association between AID, PED, and CI. We excluded 503 cases of missing dietary index data, 417 cases of missing CI data, as well as 4560 individuals with missing covariate data. After eliminating the merged missing values, the final analysis sample utilized in this study contained 8692 adults aged 60 years or older. The specific data cleaning process is shown in Figure 1. All subjects signed informed consent for baseline and follow-up surveys. The project was approved by the Biomedical Ethics Committee of Peking University, China (IRB00001052-13074).

### 2.2. Assessment of Dietary Index

Five protein-enrich food sources were considered in this research, which are meats, fish, eggs, dairy and its products, and bean products. We constructed the PED, which ranged from 0 to 5, by summing up the frequency of intake of protein-enriched food. The details of protein-enriched foods have been described in an earlier study [24]. What is more, the anti-inflammatory food included vegetables, fruits, legumes and their products, nuts, and tea, and they were used to establish the AID, ranging from 0 to 5. The consumption of any food category above “frequently or virtually every day” was deemed to equal one AID unit [31], and the AID scores are obtained by summing the scores of each food consumed.

### 2.3. Assessment of Cognitive Impairment

The CI of CLHLS participants was assessed by the Chinese version of the Mini-Mental State Examination (MMSE) through a home-based interview, which includes 24 items, covering 7 subscales including orientation (4 points for time orientation and 1 point for location orientation); naming foods (naming as many kinds of food as possible in 1 min, 7 points); registration of 3 words (3 points); attention and calculation (mentally subtracting 3 iteratively from 20, 5 points); copying a figure (1 point); recall (delayed recall of the 3 words mentioned above, 3 points); and language (2 points for naming objectives, 1 point for repeating a sentence, and 3 points for listening and following directions). The MMSE score ranges from 0 to 30, and higher scores represent a better CF. The validity and reliability of this Chinese MMSE have been validated in previous studies [56,57]. The judgment of CI is related to the level of education received: illiteracy scores ≤ 17, primary school level scores ≤ 20, and high school or above scores ≤ 24 are considered to have CI.

### 2.4. Covariates

In order to minimize their impact, we controlled for a large number of potential confounding factors, including age, gender (male or female), residence (urban or rural), nationality (“Han” or “other”), and education level (0 year, 1–6 years, and ≥7 years). Smoking and alcohol consumption were binary variables (yes or no). The calculation method for body mass index (BMI) was to divide weight (kilograms) by the square of height (meters). In addition, we also classified marital status into four categories: unmarried, married, divorced or separated, and widowed, and assigned scores of 0, 1, 2, 3. Assigned a score of 0 to “good”, “very good”, and “average” in the self-assessment of life satisfaction and health condition, and assigned a score of 1 to “not good” and “very bad”. Finally, we categorized sleep duration into three levels: less than 7 h, 7–9 h, and more than 9 h, with scores of 0, 1, 2, respectively.

### 2.5. Statistical Analysis

Continuous variables that follow a normal distribution are represented by mean ± standard deviation (SD). Categorical variables are represented in terms of frequency and percentage. Analysis of variance or Chi-square test is used to compare differences among subjects under different demographic characteristics. We use a binary logistic regression model to evaluate the association of CI with PED and AID separately. Four hierarchical regression models have been established: the basic model (Model 1) does not include any covariate; Model 2 controlled age, gender, health condition, life satisfaction, and sleep duration; Model 3 further controlled for smoking, drinking, and marital status; Model 4 additionally controlled BMI, nationality, and residence.

Restricted cubic splines (RCS) are adept at handling the non-linear relationship between continuous variables and response variables. We used RCS to determine whether there is a non-linear relationship between AID, PED, and CI. Additionally, we performed subgroup analysis for the decline of CF, grouped by gender, residence, sleep duration, health condition, and life satisfaction, and tested for interactions between grouping variables and two dietary patterns using likelihood ratio tests. We also adopted sensitivity analyses using the full model (Model 4) under different participants. All statistical analyses were completed using SPSS 26.0 and R 4.3.0. *p* < 0.05 indicates statistical significance in this study.

## 3. Results

### 3.1. The Characteristics of Study Participants

In this study, we used information from 8692 participants, whose mean age was 83.53 ± 11.48. The gender ratio was relatively balanced (44.8% male and 55.2% female), and their nationality was mostly Han (95.1%). More than half of the population had a BMI value between 18.5 and 23.9 (54.5%). In addition, the vast majority of them lived in rural areas (82.6%) and had a relatively low level of education, with only 20.6% receiving more than six years of education. Among them, 15.9% smoked, 14.9% drank alcohol, and more than half of the older adults slept less than 7 h (53.2%). They had a good self-evaluation of life satisfaction and health condition, accounting for 97% and 86.5%, respectively. The AID, PED, and CI scores were significantly different among these subgroups: BMI, gender, education, marital status, drinking, sleep duration, and health condition. More detailed information can be found in Table 1.

### 3.2. Association between AID, PED, and CI

In this study, we found that among older adults in China, the higher the AID (OR = 0.694, 95% confidence interval: 0.657–0.734, *p* < 0.001) and PED (OR = 0.902, 95% confidence interval: 0.864–0.941, *p* < 0.001) dietary scores, the lower the odds of suffering from CI in Model 1 without controlling any covariates. After controlling some demographic variables (age, gender, health condition, life satisfaction, sleep duration, smoking, drinking, marital status, BMI, nationality, and residence) hierarchically in Model 2 to Model 4, the association was reduced, but remained statistically significant. Moreover, in the full model (Model 4), taking into account all the relevant covariates, AID (OR = 0.789, 95% confidence interval: 0.740–0.842, *p* < 0.001) and PED (OR = 0.910, 95% confidence interval: 0.866–0.956, *p* < 0.001) were still related with a reduced risk of suffering from CI, by 21% and 9%, respectively. More detailed information is provided in Table 2.

### 3.3. Restricted Cubic Splines in the Regression Model

RCS did not reveal any significant non-linear relationship between two dietary patterns and CI in the full model (AID: *p*_overall_ < 0.001, *p*_non-linear_ = 0.063; PED: *p*_overall_ = 0.001, *p*_non-linear_ = 0.421). Figure 2 shows the risk of CI decreased when AID or PED increased. We can see that in Part A, the risk of CI decreases rapidly with the increase in the AID scores before the OR value drops to 1. As can be seen from Part B, the CI risk also shows a decreasing trend with the increase in PED scores, but its overall trend is relatively gentle compared to AID. Furthermore, we found that the protective effect on CI was significantly increased when the intake dose of both diets exceeded 3 units, namely that it is recommended to eat more than three types of protein-enrich foods and anti-inflammatory foods regularly every day to prevent CI.

### 3.4. Subgroup Analysis

Figure 3 and Figure 4 present the results of subgroup analysis. Based on subgroup analysis, we found that these associations are robust in subgroups of CI risk variables such as gender, residence, sleep duration, health condition, and life satisfaction. In addition, statistically significant interactions between residence, sleep duration, health condition, life satisfaction, and two dietary patterns (AID, and PED) on CI were observed (*p*-Interaction < 0.01). We can see from the figure that both dietary patterns still have a protective effect on CI under different subgroups, with all OR values below 1 and *p* < 0.001.

### 3.5. Sensitivity Analysis

We performed sensitivity analysis using the full model (Model 4) under different participants. All of the associations between AID, PED, and CI are significant and consistent with the main results in this research. Exclusion criteria for sensitivity analysis: (1) excluded the older adults who sleep less than 7 h every day; (2) excluded the participants with poor health condition.

## 4. Discussion

The findings of this study are based on a nationwide population of older adults in China. We found that older adults with higher AID and PED scores are less likely to suffer from CI. Furthermore, we used RCS to determine whether there was a non-linear relationship between these two dietary patterns and CI, and demonstrated the robustness and reliability of our research results through subgroup analysis and sensitivity analysis.

Previous research has confirmed a number of risk factors that can impair normal CF, such as advanced age, lower levels of education, and unhealthy lifestyles, as well as physical and mental health disorders [58]. Our results are consistent with previous studies; people who have received education for a longer period of time have higher MMSE scores, which also confirms they have better CF due to the longer education. We found that a difference in CI exists among females and males (*p* < 0.001), similar to many previous studies [59,60]. This may be due to the previous differences in socio-economic status between men and women in society, which resulted in fewer opportunities for women to receive education, leading to older women becoming a more vulnerable group to CI [57]. In this study, there is also a significant gender difference in the scores of the two dietary patterns; studies have proved significant differences in dietary intake between men and women [61]. Therefore, one potential reason for this difference in the relationship between dietary patterns and CF may be gender [62]. Based on our exploration of the impact of dietary patterns on CI, it is reasonable that gender differences in dietary intake can lead to gender differences in CI. Moreover, we found a significant difference in CI scores with smoking, which is consistent with previous studies; current smoking may be positively correlated with CI in middle-aged people [63]. In older women, habitual sleep duration predicted future risk of CI, including dementia, independent of vascular risk factors [64], and individuals with insufficient sleep time (≤4 h per night) or excessive sleep time (≥10 h per night) may have an impact on CF [65,66]. Those explain why there were significant differences in CI by gender and sleep duration in this study. An Indian study found that mental health disorders such as CI and depressive symptoms in older adults differed in rural/urban areas [67], perhaps due to differences in socio-economic and educational levels between rural and urban areas. Additionally, our research findings have indicated that married older adults had significantly higher CF scores than other groups. Marriage is associated with a lower likelihood of CI; it provides social support, companionship, and participation in mental stimulation activities, which can help improve cognitive health [68]. Conversely, divorced and widowed older adults are particularly susceptible to CI [69]. In this study, individuals with better self-evaluation of life satisfaction and health condition had better CF. Satisfaction with life may be a positive psychological resource for maintaining CF and preventing the risk of dementia [70]. Some studies have found that sub-health condition may be a risk factor for CI in northern Chinese population. Early screening of sub-healthy individuals, as well as emergency treatment of sub-healthy individuals, may contribute to the prevention of CI [71].

Our research findings support evidence from a previous study on older adults that nutrition can alter the risk of future CI and dementia [72]. Some inconclusive evidence exists (mainly from observational studies and rarely from clinical trials) indicating a protective association between certain nutrients (such as folate, flavonoids, vitamin D, and certain lipids) or food groups (such as seafood, vegetables, and fruits, as well as potential moderate alcohol and caffeine intake) and cognitive outcomes in older adults. Various elements of the diet may be linked to CI and dementia; research has shown that a higher intake of monounsaturated fats can prevent mild CI in males and females aged 60–64 [73], and in people aged 50 years, intake of polyunsaturated fats was associated with a reduced risk of dementia [74]. A meta-analysis found a negative dose–response relationship between serum vitamin D concentration and the risk of developing dementia or Alzheimer’s disease [75]. There are studies reporting that consuming fish has an overall protective effect on cognitive decline [76]; this may be because fish products are rich in creatine. Another study found that among participants aged 60 and above, those who frequently consume milk and dairy products have a reduced risk of developing Alzheimer’s disease [77]. Fish, milk, and dairy products are all protein-enrich, which indirectly proves that studying PED has positive implications for CF. Although studies have shown that animal and plant-based proteins have a protective effect on CI [78], there is no specific study on the association between PED diets and CI among the population of older adults in China.

The incidence of CI in the elderly seems to be decreasing, indicating that the cohort effect of lifestyle factors is playing a role, and diet may also be a promising strategy for delaying, slowing down, or preventing CI [79,80,81,82]. A study has found that Mediterranean and DASH diets are associated with cognitive outcomes, such as lower incidence of CI and cognitive decline, or lower risk of Alzheimer’s disease [83]. Inflammation is an important mechanism of cognitive dysfunction. The Systemic Immune Inflammatory Index and Systemic Inflammatory Response Index are two blood inflammatory markers associated with many chronic disorders, including CI. Cohort studies have shown that pro-inflammatory diets are significantly associated with CI [84]. Korean cross-sectional studies have shown that DII scores are negatively correlated with overall CF and verbal memory [85], and higher DII scores are associated with CI in women aged 65 to 79 [86]. Anti-inflammatory foods include vegetables, fruits, legumes and their products, nuts, and tea. The above dietary studies that are beneficial for CF include some anti-inflammatory foods. Consequently, it is of great significance for us to systematically construct AID and explore the relationship between AID and CI. Our study found that AID and PED patterns have a protective effect on CI; the higher the score of the two dietary patterns, the lower the probability of CI, which is very meaningful for improving and preventing CI in older adults from a dietary perspective in the future.

In addition to diet, malabsorption and some digestive problems may also be related to CI; a study has found that malabsorption of vitamin B-12 can have an impact on cognitive performance [87], and some intestinal disorders, such as celiac disorder, can also impair CF [88,89]. Furthermore, due to reduced dietary intake and poor intestinal malabsorption, the incidence of folate deficiency is high in people aged ≥65 years old, and population-based studies have shown that low folate levels are associated with mild CI, dementia (especially Alzheimer’s disease), and depression [90]. Our research found that when the consumption of PED and AID was greater than 3 units, their protective effect on CI was more significant. That is to say, it is recommended to frequently eat more than 3 protein-rich foods and anti-inflammatory foods every day to prevent CI. According to the linear relationship proven by RCS, the greater the consumption of the two dietary patterns, the less likely they are to suffer from CI. However, considering the issues of digestion and absorption, we cannot excessively consume these diets to prevent CI. Therefore, in order to determine the more reasonable intake doses of two diets that can prevent CI while avoiding the problem of malabsorption, it is necessary to perform further exploration through more professional research on diet and digestion in future studies.

## 5. Limitations

To the best of our knowledge, this is the first study to systematically construct a protein-enriched diet index and anti-inflammatory diet index and explore the relationship between them and CI. However, some limitations requiring improvement still exist. Firstly, the data on CI are self-reported and may have some bias. Secondly, although we further confirmed the reliability and robustness of the research results using subgroup analysis and sensitivity analysis, cross-sectional studies are not sufficient to infer causal relationships. Therefore, in subsequent research, we could confirm the possible causal relationship between them through cohort studies. Last but not least, although we found that when the consumption of PED and AID exceeds 3 units, their protective effect on CI is more significant (owing to the linear relationship between the two dietary patterns and CI), the optimal dosage for achieving the best effect remains to be determined. Therefore, the precise dosage recommendations for the two dietary patterns need to be further demonstrated in future cohort studies.

## 6. Conclusions

In this study, we discovered that higher intakes of AID and PED are associated with a lower risk of CI among older adults in China, there is a linear relationship between them, and the main results are also significant under different subgroups. These findings have important implications for prevention and control of Cl from a dietary and nutritional perspective in the future.

## Figures and Tables

**Figure 1 nutrients-16-01333-f001:**
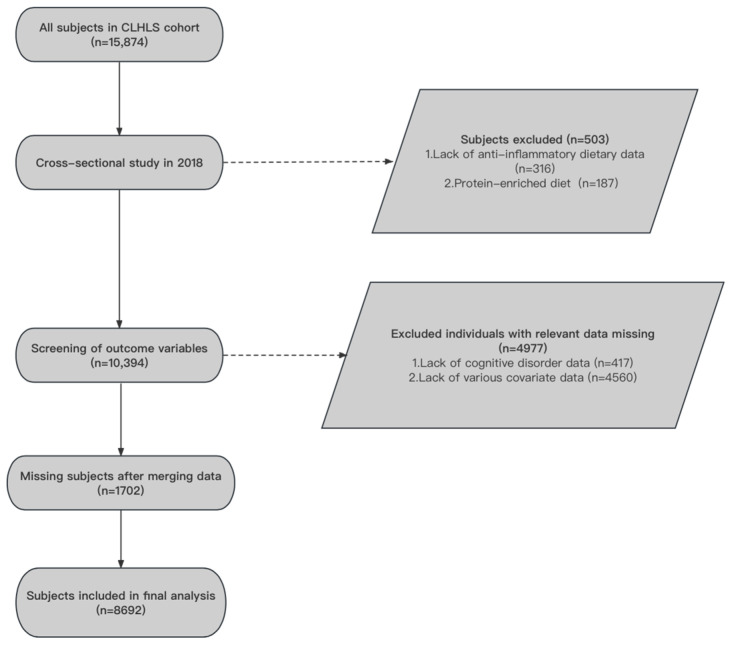
Data cleaning flow chart.

**Figure 2 nutrients-16-01333-f002:**
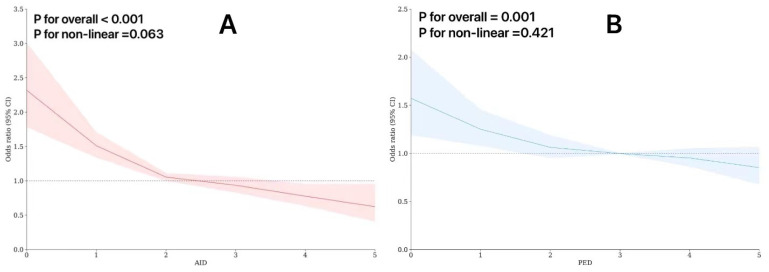
Restricted cubic spline for testing the hypothesis of non-linear correlation between AID, PED, and CI. (**A**) The linear relationship between AID and CI; (**B**) the linear relationship between PED and CI.

**Figure 3 nutrients-16-01333-f003:**
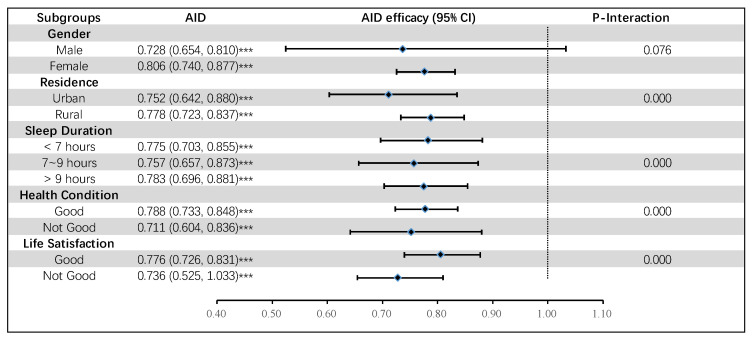
Associations of AID with CI among subpopulations. Notes: AID: anti-inflammatory diet, *** *p* < 0.001.

**Figure 4 nutrients-16-01333-f004:**
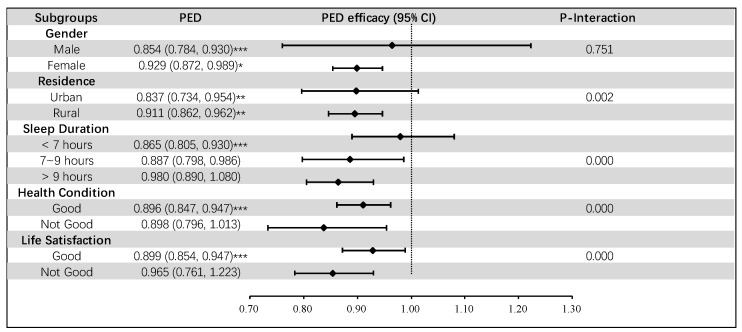
Associations of PED with CI among subpopulations. Notes: PED: protein-enriched diet, *** *p* < 0.001, ** *p* < 0.01, * *p* < 0.05.

**Table 1 nutrients-16-01333-t001:** Basic demographic characteristics of the participants.

Variable	Frequency (%)	AID (M ± SD)	*p* Value	PED (M ± SD)	*p* Value	CI (M ± SD)	*p* Value
Age, years	8692 (100%)	2.32 ± 1.18	*p* < 0.001	2.95 ± 1.44	0.093	25.28 ± 6.01	*p* < 0.001
BMI, kg/m^2^							
<18.5	1304 (15%)	2.05 ± 1.12	*p* < 0.001	2.82 ± 1.38	*p* < 0.001	22.65 ± 7.42	*p* < 0.001
18.5–23.9	4740 (54.5%)	2.27 ± 1.18		2.91 ± 1.45		25.25 ± 5.97	
23.9–27.9	2145 (24.7%)	2.54 ± 1.18		3.11 ± 1.43		26.68 ± 4.80	
>27.9	503 (5.8%)	2.49 ± 1.15		3.01 ± 1.45		26.46 ± 4.57	
Gender							
Male	3896 (44.8%)	2.47 ± 1.22	*p* < 0.001	3.08 ± 1.39	*p* < 0.001	26.54 ± 4.97	*p* < 0.001
Female	4796 (55.2%)	2.19 ± 1.14		2.84 ± 1.46		24.27 ± 6.57	
Nationality							
Han	8262 (95.1%)	2.33 ± 1.19	*p* < 0.001	2.98 ± 1.44	0.024	25.30 ± 5.99	0.066
Others	430 (4.9%)	2.03 ± 1.08		2.45 ± 1.27		24.92 ± 6.52	
Education (years)							
0	3940 (45.3%)	1.98 ± 1.06	*p* < 0.001	2.65 ± 1.43	*p* < 0.001	22.78 ± 6.87	*p* < 0.001
1–6	2965 (34.1%)	2.34 ± 1.14		2.94 ± 1.38		26.94 ± 4.37	
>6	1787 (20.6%)	3.03 ± 1.18		3.63 ± 1.31		28.07 ± 3.73	
Marital status							
Unmarried	68 (0.8%)	1.93 ± 1.14	*p* < 0.001	2.46 ± 1.49	*p* < 0.001	25.32 ± 5.18	*p* < 0.001
Married	3854 (44.3%)	2.51 ± 1.21		3.05 ± 1.42		27.53 ± 3.74	
Divorce or Separation	184 (2.1%)	2.47 ± 1.21		3.17 ± 1.41		26.88 ± 4.50	
Widowed	4586 (52.8%)	2.16 ± 1.13		2.87 ± 1.45		23.33 ± 6.89	
Smoking							
Yes	1379 (15.9%)	2.40 ± 1.19	0.275	2.94 ± 1.38	0.017	26.60 ± 4.83	*p* < 0.001
No	7313 (84.1%)	2.30 ± 1.18		2.95 ± 1.45		25.04 ± 6.18	
Drinking							
Yes	1292 (14.9%)	2.56 ± 1.23	*p* < 0.001	3.11 ± 1.37	0.007	26.58 ± 4.81	*p* < 0.001
No	7400 (85.1%)	2.28 ± 1.17		2.92 ± 1.45		25.06 ± 6.17	
Sleep Duration (h)							
<7 h	4628 (53.2%)	2.31 ± 1.20	*p* < 0.001	2.91 ± 1.48	0.014	25.71 ± 5.58	*p* < 0.001
7~9 h	2451 (28.2%)	2.40 ± 1.16		3.02 ± 1.40		26.04 ± 5.40	
>9 h	1613 (18.6%)	2.22 ± 1.14		2.97 ± 1.34		22.91 ± 7.36	
Life Satisfaction							
Good	8431 (97%)	2.34 ± 1.18	0.096	2.98 ± 1.43	0.202	25.39 ± 5.93	*p* < 0.001
Not Good	261 (3%)	1.68 ± 1.10		2.19 ± 1.50		22.01 ± 7.60	
Residence							
Urban	1512 (17.4%)	2.95 ± 1.19	0.485	3.71 ± 1.31	*p* < 0.001	26.58 ± 5.37	*p* < 0.001
Rural	7180 (82.6%)	2.19 ± 1.14		2.79 ± 1.41		25.01 ± 6.11	
Health Condition							
Good	7516 (86.5%)	2.37 ± 1.18	*p* < 0.001	3.00 ± 1.42	*p* < 0.001	25.52 ± 5.83	*p* < 0.001
Not Good	1176 (13.5%)	2.00 ± 1.16		2.67 ± 1.50		23.78 ± 6.90	

Notes: AID: anti-inflammatory diet, PED: protein-enriched diet, CI: cognitive impairment, SD: standard deviation.

**Table 2 nutrients-16-01333-t002:** Associations of AID, PED with CI among Chinese older adults.

Model	AID	PED
Model 1	0.694 (0.657, 0.734) ***	0.902 (0.864, 0.941) ***
Model 2	0.795 (0.746, 0.846) ***	0.914 (0.871, 0.960) ***
Model 3	0.798 (0.749, 0.850) ***	0.917 (0.874, 0.962) ***
Model 4	0.789 (0.740, 0.842) ***	0.910 (0.866, 0.956) ***

Notes: AID: anti-inflammatory diet, PED: protein-enriched diet, *** *p* < 0.001.

## Data Availability

The CLHLS data are available at https://opendata.pku.edu.cn/dataverse/CHADS (accessed on 3 February 2024).

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
