# Peer review of "Anti-Inflammatory Diet and Protein-Enriched Diet Can Reduce the Risk of Cognitive Impairment among Older Adults: A Nationwide Cross-Sectional Research"

_nutrients, 2024, doi:10.3390/nu16091333_

Round 1

Reviewer 1 Report

Comments and Suggestions for Authors

The authors present a very interesting topic since, not only in China but throughout the world, the average age and life expectancy are increasing.

However, some points should be highlighted in particular in the limitations of this study:

- Among the nutrients, creatine (10.2174/0118761429272915231122112748) should undoubtedly be included, which can be related to protein sources of animal origin such as meat and fish but not to milk, eggs, and beans, and beta-carotene together (10.3390/brainsci13101468) with polyphenols; omega3 fatty acids (10.3390/nu13041074) should also at least be mentioned

- Physical activity was not taken into consideration. Still, it was seen that this has a direct effect not only on the quality of life in general but also on cognitive ability (10.3389/fspor.2022.950949, 10.1016/j.arr.2022.101591)

- Although accurate, the method used does not directly analyze eating habits and does not specify for how many days such habits are tracked

A subparagraph should be added with the limitations, as they are multiple

In any case, the result has a logical and scientific rationale; a suggestion from the authors on dosages of proteins, antioxidants, and anti-inflammatories would also be interesting molecules.

Comments on the Quality of English Language

It needs revision.

Author Response

Thank you very much for your time and effort in reviewing our article! The response letter has been uploaded. Please see the attachment.

Reviewer 2 Report

Comments and Suggestions for Authors

The authors reported a nationwide cross-sectional study on the association between older persons' risk of cognitive impairment and anti-inflammatory and protein-enriched diets. 8,692 people in all were included in the analysis using data from the 2018 waves of the China Longitudinal Health and Longevity Survey. The findings indicated that a decreased incidence of cognitive impairment in older persons is linked to increased intakes of anti-inflammatory and protein-enriched foods.

The paper is interesting and well written. The function that malabsorption plays in cognition is one that I would want to draw attention to (10.1016/B978-0-7020-4087-0.00042-5; 10.1097/MCG.0b013e318159c654; 10.1007/s13760-017-0870-z) please consider in your discussion.

Author Response

(The authors gave the same response as above.)

Round 2

Reviewer 1 Report

Comments and Suggestions for Authors

The authors improved enough the manuscript.

Comments on the Quality of English Language

It was improved too, just some to review

Author Response

(The authors gave the same response as above.)
